

# Curcumin Chalcone Derivatives Database (CCDD): a Python framework for natural compound derivatives database

Shailima Rampogu[1],*, Thananjeyan Balasubramaniyam[2],* and Joon-Hwa Lee[2]

[1] Cachet Big Data Labs, Hyderabad, Telangana, India
[2] Department of Chemistry, Gyeongsang National University, Jinju, Gyeongnam, South Korea
* These authors contributed equally to this work.

## ABSTRACT

We built the Curcumin Chalcone Derivatives Database (CCDD) to enable the effective virtual screening of highly potent curcumin and its analogs. The two-dimensional (2D) structures were drawn using the ChemBioOffice package and converted to 3D structures using Discovery Studio Visualizer V 2021 (DS). The database was built using different Python modules. For the 3D structures, different Python packages were used to obtain the data frame of compounds. This framework is also used to visualize the compounds. The webserver enables the users to screen the compounds according to Lipinski's rule of five. The structures can be downloaded in .sdf and .mol format. The data frame (df) can be downloaded in .csv format. Our webserver can help computational drug discovery researchers find new therapeutics and build new webservers. The CCDD is freely available at: https://srampogu-ccdd-ccdd-8uldk8.streamlit.app/.

## INTRODUCTION

Modern drug development involves identifying possible hits and optimizing them to improve selectivity, specificity, activity, and extent of absorption. Once a molecule with all of the necessary qualities has been found, drug development and clinical trials can begin. Computer-aided drug design is used in one or more phases of the drug discovery procedure. The pharmaceutical business invests a significant amount of money in the modern drug discovery process.

Natural compounds and their derivatives have been at the forefront in contributing to the therapeutic field. The compounds that have gained enormous credit are curcumin and chalcone.

Curcumin is a diarylheptanoid polyphenol with two methoxy rings and an ortho phenolic OH group. Curcumin is attached to a conjugated seven-carbon chain with an enone and 1,3-diketone group. Curcumin has two double bonds, o-methoxy and phenolic groups, and a 1,3-keto-enol (Fig. 1A) (*Yang et al., 2017*). Chalcones are a class of

Corresponding author
Joon-Hwa Lee, joonhwa@gnu.ac.kr

**(A)** **(B)**

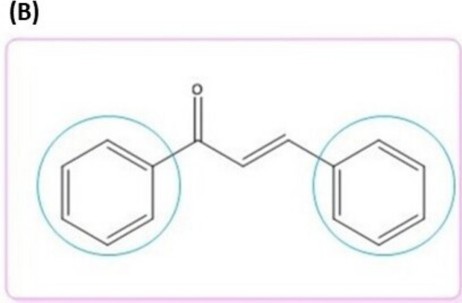

**Figure 1 Molecular structures of (A) curcumin and (B) chalcone.** The blue circle denotes the possibility for synthesizing new derivatives.

open-chain flavonoid compounds with a carbonyl group composed of two aromatic rings and a range of α,β-unsaturated substituents (Fig. 1B) (*Salehi et al., 2021*).

The rhizome of the turmeric plant, *Curcuma longa*, contains the polyphenol curcumin, also known as diferuloylmethane, which is also present in other *Curcuma* species and is widely used as a medicinal herb in Asian countries (*Hewlings & Kalman, 2017*). Traditional treatments have employed curcumin as a home remedy to treat a number of illnesses, including rheumatism, sinusitis, anorexia, cough, hepatic disorders, and biliary disorders (*Asakura & Kitahora, 2018*). Curcumin also acts against diabetes, multiple sclerosis, cardiovascular diseases, inflammation, and Alzheimer's disease (*Suresh, Yadav & Suresh, 2006*; *Fadus et al., 2017*). Due to curcumin's high therapeutic potential, several research groups have developed analogs that have proven to be active compounds against several diseases (Fig. 2).

One elite class of natural compounds is chalcone, which is a good source of therapeutics (*Ni, Meng & Sikorski, 2004*; *Katsori & Hadjipavlou-Litina, 2011*; *Salehi et al., 2021*). Chalcone is bountiful in vegetables such as bean sprouts, tomatoes, potatoes, shallots, fruits such as apples, citruses, tea, spices, and various plants (*Orlikova et al., 2011*; *Zhuang et al., 2017*). The term chalcone is derived from the Greek word *chalcos* which means bronze (*Zhuang et al., 2017*; *Janse van Rensburg, Legoabe & Terre'Blanche, 2021*; *Salehi et al., 2021*). The term chalcones was coined by Kostanecki and Tambor (*Tekale, 2020*).

The other names of chalcones are β-phenylacrylophenone, α-phenyl-β-benzoylethylene, benzylideneacetophenone, and phenyl styryl ketone (*Tekale, 2020*). Generally, chalcones are benzyl acetophenone compounds consisting of two fragrant rings A and B joined by three aliphatic carbons, and they have α,β-unsaturated ketones demonstrating varied organization of substituents (*Salehi et al., 2021*). The biological actions of chalcones, which include antibacterial, anticancer, cytotoxic, antioxidative, anti-inflammatory, and antiviral properties, are well characterized. Chalcone derivatives are being utilized extensively in the treatment of viral infections, cardiovascular conditions, stomach cancer, food additives, and chemicals in cosmetic formulations (Fig. 2) (*Salehi et al., 2021*).

The Natural Compound Derivatives Database (NCDD) objectives include establishing a database containing an extensive number of derivative compounds in a single location, as

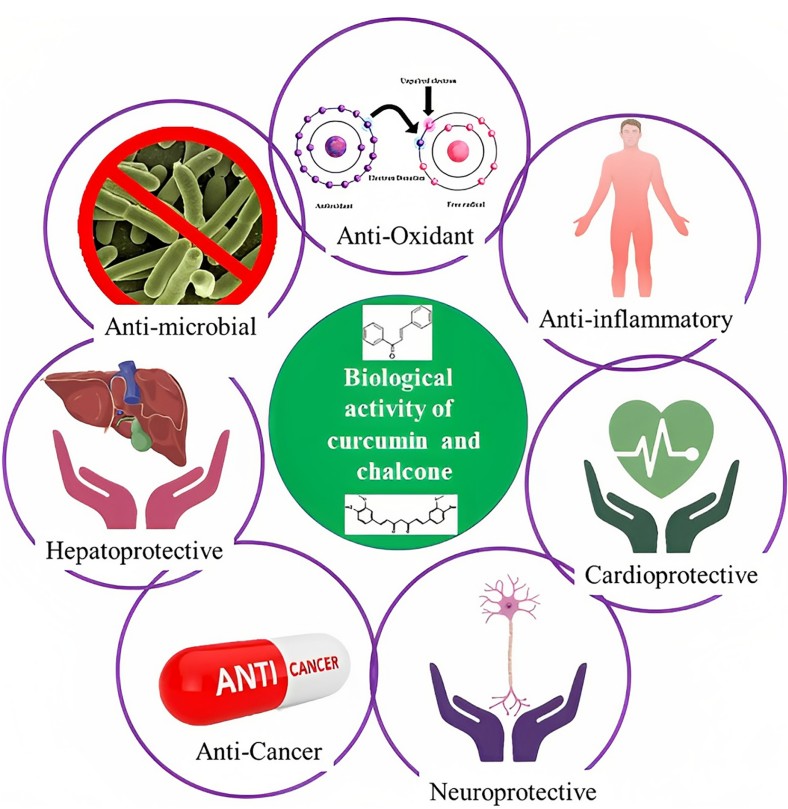

**Figure 2 Schematic illustration of curcumin and chalcone biological activities.**

there is currently no suitable web platform that can provide users with analog compounds in a single location in the Curcumin Chalcone Derivatives Database (CCDD) (*Gallo et al., 2023*).

To expedite the drug design process, computational methodologies are advantageous for interpreting and guiding experiments (*Walsh, 2003*). The two main categories of computer-aided drug design (CADD) methods are structure-based drug design (SBDD) and ligand-based drug design (LBDD) (*Schneider & Fechner, 2005*). Utilization of open access resources for cheminformatics and structural bioinformatics as well as public platforms for code deposition, such as GitHub, is increasing in the realm of research. This combination facilitates and encourages the development of modular, reproducible, and easy-to-share computer-aided drug design (CADD) pipelines (*Pirhadi, Sunseri & Koes, 2016*).

Virtual screening is an essential step in computational drug discovery (CDD). CDD approaches reduce the time of discovering a new drug candidate and involve lower costs and human burden. Typically, CDD consists of the screening of databases of small molecules either by pharmacophore modeling methods or molecular docking approaches. The small molecules from the databases will be subjected to Lipinski's rule of 5 to enhance the opportunities of the candidate drug during clinical trials.
## METHODS

### Data curation

We manually extracted information on the structures of chalcone and curcumin from the literature. With the use of keywords such as "curcumin", "curcumin derivatives", "modified curcumin", "chalcone", "modified chalcone", and "chalcone derivatives", we searched the PubMed database and Google Scholar. More than 30 complete literary works were downloaded overall. To learn more about the specifics of curcumin and chalcone structures, we manually searched the literature.

To build this webserver, we used different modules from Python. The structures of curcumin and chalcone analogs were retrieved from the literature. The curcumin analogs were obtained from a literature search (*Pabon, 1964*; *Ohtsu et al., 2003*; *Robinson et al., 2003*, *2005*; *Nichols et al., 2006*; *Ohori et al., 2006*; *Simoni et al., 2008*; *Basile et al., 2009*; *Xiao et al., 2010*; *Kudo et al., 2011*; *Sreenivasan et al., 2013*; *Ahn et al., 2014*; *Rampogu et al., 2022*). The chalcone analogs/derivatives were retrieved from several literature sources (*Boeck et al., 2006*; *Choudhary & Juyal, 2011*; *Gupta & Jain, 2015*; *Bui et al., 2016*; *Pesaran Seiied Bonakdar et al., 2017*; *Gomes et al., 2017*; *Teng et al., 2018*; *Zhu et al., 2019*; *Higgs et al., 2019*; *Kaur, Singh & Narasimhan, 2019*; *Aljamali, Hamzah Daylee & Jaber Kadhium, 2020*; *Rammohan et al., 2020*; *Okolo et al., 2021*; *Vadivoo et al., 2021*; *Wijayanti et al., 2021*; *Konidala et al., 2021*; *Abosalim, Nael & El-Moselhy, 2021*; *Ouyang et al., 2021*; *Rampogu et al., 2021*; *Taresh, 2022*) to build a webserver that could help in the virtual screening process.

### Database generation

Chemdraw professional software was employed to draw the two-dimensional (2D) structure of the derivatives, which was then uploaded to the Discovery Studio (DS) visualizer 2021 to obtain the 3D structures. These structures were saved in the .sdf format to the local disk and then uploaded to Jupyter Notebook. The dataframe (df) was then generated using *PandasTools*. Correspondingly, the unused columns were deleted and Lipinski's rule of five was calculated for df and stored in the .csv file. This .csv file was loaded onto *Streamlit* to create the webserver (Fig. 3). The code was written in Python IDLE Shell 3.9.5 (*Napoles-Duarte et al., 2022*; *Cihan Sorkun et al., 2022*; *Karade & Karade, 2023*).

## RESULTS

The CCDD was built using the .csv file obtained from Python and can be divided into three steps. In the first step, the 3D structures from the DS were upgraded onto the Jupyter Notebook, and the data frame was created using rdkit and *PandasTools*. Then, Lipinski's descriptors were then calculated, and all the data with null values and duplicates were deleted. The resultant 474 compounds were saved in the .csv format and were taken onto *Streamlit* to build the webserver.

In the second step, the code was written in Python IDEL Shell for *Streamlit*. The overview of the framework shows the df and the filter in terms of parameters corresponding to Lipinski's rule of 5. Upon moving each parameter the changes are
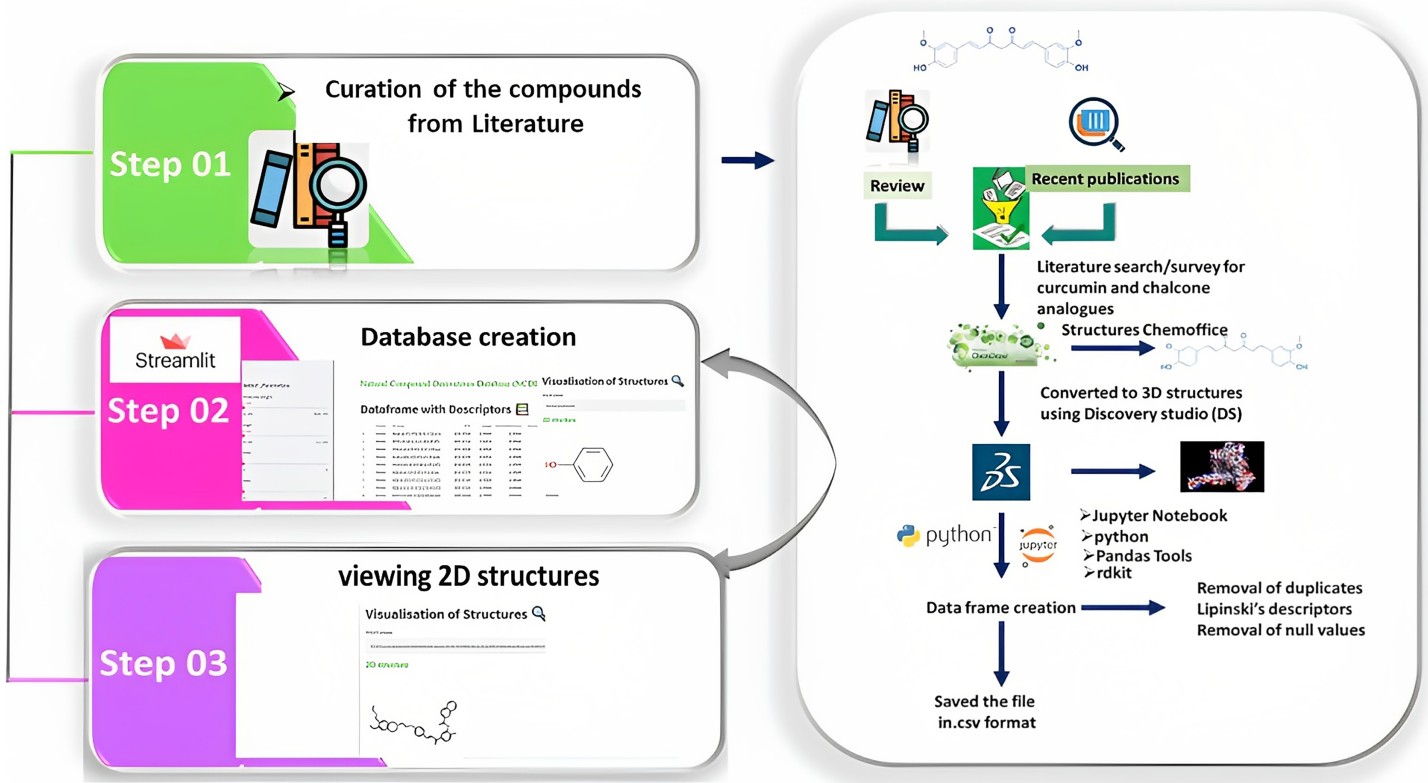

**Figure 3 Illustration of the methods adopted for building the webserver.** The curation of curcumin and chalcone derivatives is carried out in Step 1. The left section describes how to draw 2D structures and then convert them into 3D structures. This demonstrates how to prepare a dataframe and save it as the .cvs file format. Step 2 details the construction of a database using *Streamlit*, while Step 3 demonstrates the visualization of 2D structures in a webserver builtwith *Streamlit*.     

reflected in df. Users can filter the df according to their use and the df can be downloaded (in .csv format) by clicking the 'Get Compounds' button.

The third step is to view the structure on the DS, form the .csv file, delete all the contents and retain only the SMILES and save the file. Then, the file format should be changed from .csv to .smi using the rename option. Then drag and drop the .smi file onto the DS. The filtered data can be viewed in 3D structure form.

The 2D structures can be viewed by entering the SMILES code of the selected compound and then clicking *enter*. Furthermore, the database is equipped to retrieve the data in the row-wise selection after entering the SMILES.

**How to access CCDD: access the database of curcumin chalcone derivatives. Typically, a database consists of four layers**

(1) The df has the SMILES code of compounds along with their corresponding descriptors. The side bar panel is equipped with the app to screen the compounds according to the user's interest (Fig. 4). The selected compounds or the df in whole can be downloaded in the .csv format by clicking '*Get compounds*' (Fig. 4).
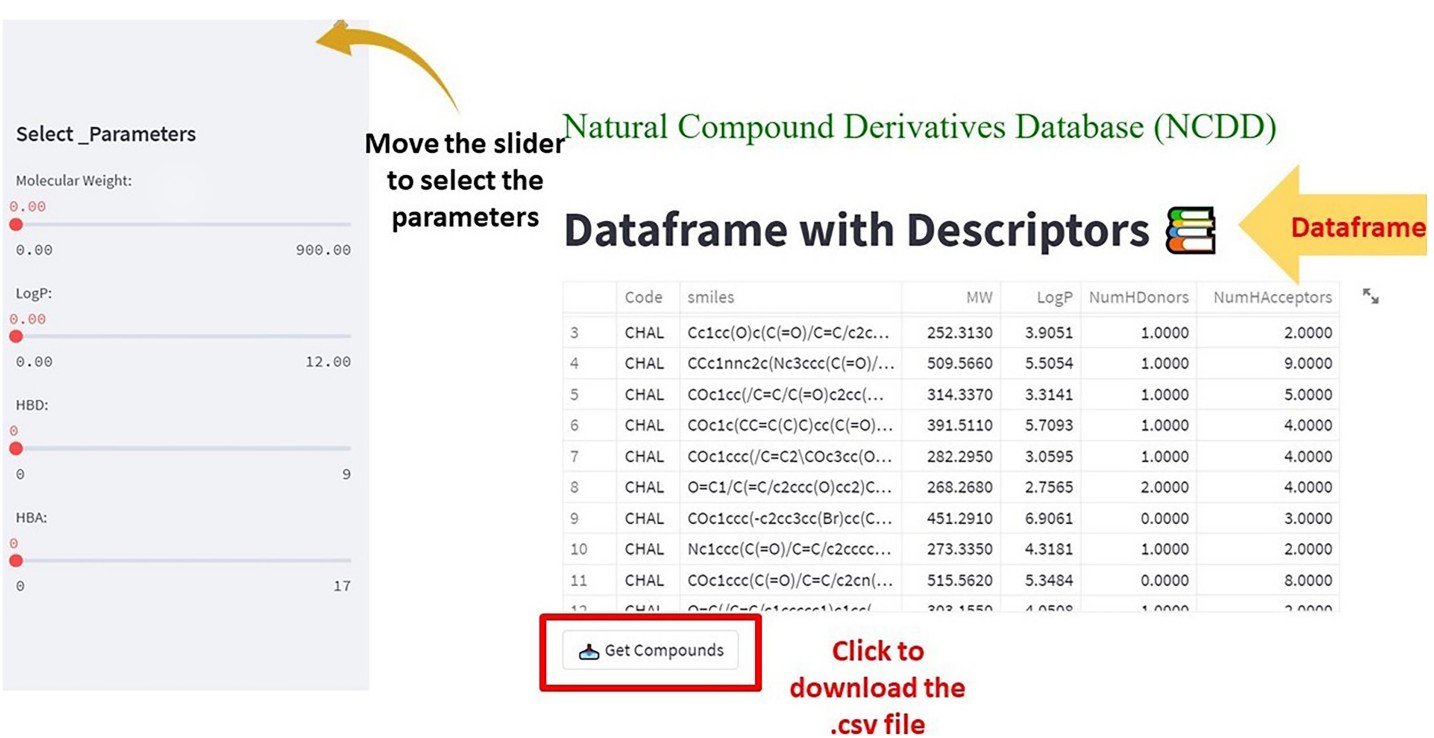

**Figure 4** The web app showing the dataframe and the parameters to select the compounds from the dataframe. The 'Get Compounds' is used to download the selected dataframe.                                                                           

(2) After opening the .csv file, delete all columns, retaining the SMILES column. Save the file to the .smi format. Drag and drop it on the DS visualizer to the view of 3D structures (Fig. 5).

(3) Within the web app, the 2D structure can be viewed by giving the SMILES as an input (Fig. 6A).

(4) Furthermore, a particular row can also be retrieved by giving the input as SMILES (Fig. 6B).

## UTILITY AND DISCUSSION

The CCDD is a free resource for scientists and research groups studying natural products. The database comprises curcumin and chalcone analogs, the largest collection of their analogs in 2022. This webserver is useful for scientists and researchers studying natural products (curcumin and chalcones) for a variety of purposes, including medication discovery and research into biodiversity. Additionally, the CCDD is the first significant chemical database with analog compounds of curcumin and chalcones that use Python frame work and *streamlit*.

This server is built using different python-based tools from the 2D structures. The Jupyter Nootbook and *PandasTool* were adapted to create the filterable *df*. This allows users to filter the data based on descriptors according to Lipinski's rule. Since this parameter is already computed, users can directly use the data according to their needs or they can simply download the whole content in .csv form.
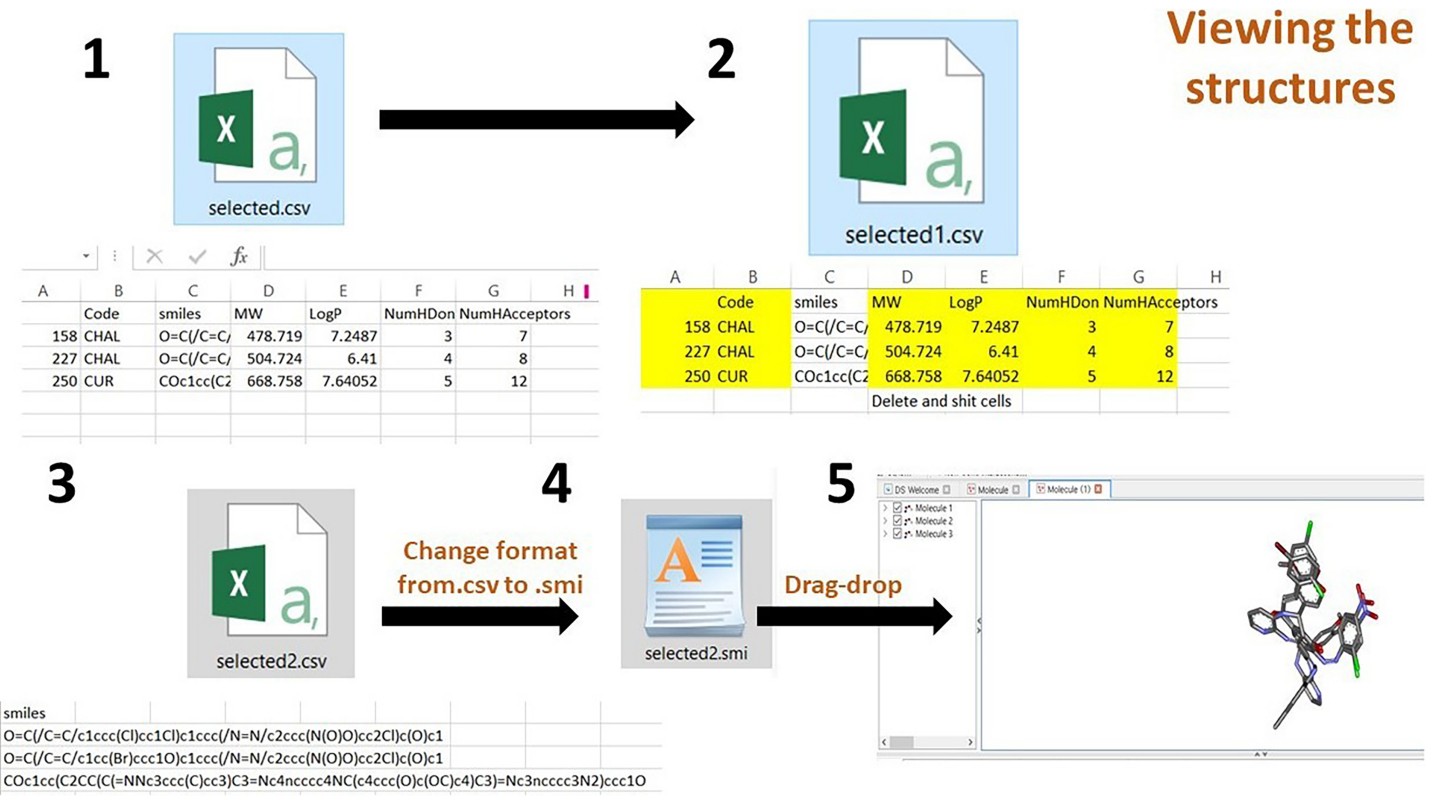

**Figure 5  Process of viewing the 3D structures after downloading the selected compounds from the dataframe.**

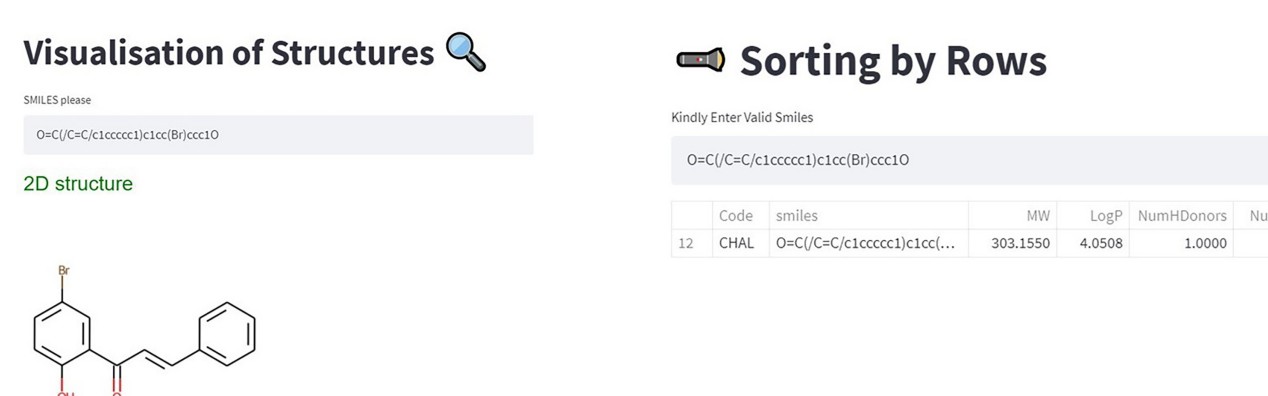

**Figure 6  (A) Visualization of the compounds and (B) retrieving the specific rows in the SMILES.** The structures can be viewed in 2D.

The database currently has 474 analogs. To the best of our knowledge this is the first web platform containing analog compounds. The database will be regularly updated to furnish new compounds.

## LIMITATIONS

Our website does not currently have the capability to automatically update newly introduced chemicals. In the next version, we will attempt to incorporate this functionality.

## CONCLUSIONS

A natural compound database was built taking the structures from the literature using Python modules. This server will help the virtual screening process that is usually conducted for the identification of candidate compounds. Our webserver could assist computational drug discovery researchers in finding new therapeutics for different diseases and could further serve in building new webservers.

### Funding

This work was supported by the National Research Foundation of Korea (2020R1A2C1006909 and 2022R1A4A1021817) and the Samsung Science and Technology Foundation (SSTF-BA1701-10). The funders had no role in study design, data collection and analysis, decision to publish, or preparation of the manuscript.

### Grant Disclosures

The following grant information was disclosed by the authors:
National Research Foundation of Korea: 2020R1A2C1006909 and 2022R1A4A1021817.
Samsung Science and Technology Foundation: SSTF-BA1701-10.

### Competing Interests

Shailima Rampogu is employed by Cachet Big Data Laboratories.

### Author Contributions

- Shailima Rampogu conceived and designed the experiments, performed the experiments, analyzed the data, prepared figures and/or tables, authored or reviewed drafts of the article, and approved the final draft.
- Thananjeyan Balasubramaniyam conceived and designed the experiments, performed the experiments, analyzed the data, prepared figures and/or tables, authored or reviewed drafts of the article, and approved the final draft.
- Joon-Hwa Lee conceived and designed the experiments, authored or reviewed drafts of the article, and approved the final draft.

### Data Availability

The CCDD is available at Streamlit:
https://srampogu-ccdd-ccdd-8uldk8.streamlit.app/.

## Supplemental Information

Supplemental information for this article can be found online at http://dx.doi.org/10.7717/peerj.15885#supplemental-information.

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
