# Peer review of "Curcumin Chalcone Derivatives Database (CCDD): a Python framework for natural compound derivatives database"

_PeerJ, doi:10.7717/peerj.15885_

## Round 0.1 · original submission · Major Revisions

I suggest authors to go through all the comments and address them in the revised version.

Reviewer 1 ·

Basic reporting

Professional English use was observed lacking in different sections which has been indicated in the annotated pdf.
Literature references are sufficient field background/context provided. However, introduction and methodology sections lack appropriate citations and have been indicated in the annotated pdf.

This article shows professional article structure, figures, and tables. Raw data shared wherever required. However, figures captions require revision which have been indicated.
Results presented are relevant to the hypothesis or the problem. However, problem should be revised as indicated in the annotated pdf file attached.

Experimental design

The article manifests the original primary research within the aims and scope of the journal.
Research question is NOT well defined though it is relevant & meaningful. It should be revised to state comprehensively how this research fills an identified knowledge gap.
Rigorous investigation performed to a high technical & ethical standard.
Methods described with sufficient detail & information to replicate. However, some areas require suitable citations to strengthen the methodology adopted.

Validity of the findings

Impact and novelty well assessed. Meaningful replication encouraged where rationale & benefit to literature is clearly stated.
All underlying data have been provided; they are robust, statistically sound, & controlled.
Conclusions are well stated, linked to original research question & limited to supporting results.

Additional comments

I commend the authors for their extensive data set, compiled over many years of detailed fieldwork. This manuscript should be revised and written in professional, unambiguous language. Introduction, methodology and discussion sections should be improved and should be improved before Acceptance.

Annotated reviews are not available for download in order to protect the identity of reviewers who chose to remain anonymous.

Reviewer 2 ·

Basic reporting

The authors can consider elaborating on this for the uninitiated readers regarding which are the other phases other than virtual screening (optimisation etc.): "Computer-aided drug design is used in one or more phases of the drug discovery procedure."

Experimental design

"From here, the 3D structures in the .sdf format were upgraded to Python (Jupyter Notebook)..."
Were the structures upgraded by adding more details? Or was it just uploaded to the software?

Validity of the findings

The authors should consider mentioning if any they feel there is any limitation to their study

Additional comments

Some minor typographical errors like

"The third step is to view the strcutures on the DS, form the .csv file delet all the contents and retain..."

"To the best of our knowledge this is the firsy web platform...."

---

## Round 0.2 · accepted · Accept

Thank you for addressing the reviewers' comments.

Reviewer 1 ·

Basic reporting

The article has been revised well by authors using professional English as indicated in the annotated pdf. Literature references are sufficient field background/context provided. Introduction and methodology sections have been revised very well by addition of citations.
This article shows professional article structure, figures, and tables. Raw data has also been provided wherever required. Figures' captions have been revised well now.
Problem statement has also been revised as indicated previously.
Results presented are relevant to the hypothesis or the problem.

Experimental design

The article manifests the original primary research within the aims and scope of the journal. Research question is well defined, relevant and meaningful. It has been stated comprehensively that how this research fills an identified knowledge gap. Rigorous investigation performed to a high technical & ethical standard. Methods described with sufficient detail & information to replicate. Appropriate citations have been added to strengthen the methodology adopted.

Validity of the findings

As analyzed previously, the validity of the findings, Impact and novelty have been observed. Meaningful replication encouraged where rationale & benefit to literature is clearly stated. All underlying data have been provided; they are robust, statistically sound, & controlled. Conclusions are well stated, linked to original research question & limited to supporting results.

Additional comments

I commend the authors for their extensive data set, compiled over many years of detailed fieldwork. This manuscript has been revised and written in professional, unambiguous language. Introduction, methodology and discussion sections have been improved as suggested previously.